# Impact of COVID-19 Lockdowns on the Activity and Mental Health of Older People in Indonesia: A Qualitative Study

**DOI:** 10.3390/ijerph192013115

**Published:** 2022-10-12

**Authors:** Nelsensius Klau Fauk, Elsa Dent, Gregorius Abanit Asa, Paul Russell Ward

**Affiliations:** 1Research Centre for Public Health, Equity and Human Flourishing (PHEHF), Torrens University Australia, Adelaide, SA 5000, Australia; 2Institute of Resource Governance and Social Change, Kupang 85227, Indonesia; 3Faculty of Medicine and Health Sciences, Krida Wacana Christian University, Jakarta 11510, Indonesia

**Keywords:** COVID-19 outbreak, lockdown protocols, individual and social activities, mental health challenges, older people, Indonesia

## Abstract

The COVID-19 pandemic has caused detrimental impacts on different population groups throughout the world. This study aimed to explore the impacts of the COVID-19 pandemic’s mandatory lockdown protocols on individual and social activities and mental health conditions of community-dwelling older people in Jakarta, Indonesia. A qualitative design using one-on-one in-depth interviews was employed to collect data from the participants (*n* = 24) who were recruited using the snowball sampling technique. Data analysis was guided by the five steps proposed in a qualitative data analysis framework, including familiarisation with the data, identification of a thematic framework, indexing the data, charting the data and mapping and interpreting the data. The findings showed that before the COVID-19 outbreak participants engaged in different kinds of regular individual and social activities. However, the COVID-19 outbreak and its mandatory lockdown protocols significantly influenced both their activities and social life, which led to social disconnection and financial difficulties for them. COVID-19 outbreak, mandatory lockdown protocols, and the disruption of individual and social activities of the participants also caused mental health challenges to them, including feelings of loneliness, loss, sadness, stress, and anger. The findings suggest that there is a need for intervention programs addressing the socio-economic and mental health impacts of the COVID-19 pandemic on older populations to help them cope with these challenges. Future studies involving large-scale older populations to comprehensively understand COVID-19 impacts on them are recommended.

## 1. Introduction

Shortly after its announcement on 7 January 2020 by the Chinese Centre for Disease Control, COVID-19 cases started to be reported in other countries [1,2]. As a response to the rapid spread of the virus, the World Health Organisation (WHO) announced it as a global public health emergency on 30 January 2020 [3]. As the rapid spread of the infection also led to a massively increased number of cases and death throughout the world, the WHO again announced COVID-19 as a pandemic on 30 March 2020 [4]. The current report on COVID-19 cases shows that there have been more the 529 million people infected with the virus and more than 6 million deaths globally, [5,6]. In the context of Indonesia, more than 6 million people have been diagnosed with COVID-19, with over 156,553 deaths [7].

In response to the pandemic, many countries have chosen lockdowns of varying stringency as a strategy to limit direct physical interaction among community members. These lockdowns have been effective internationally in preventing and reducing COVID-19 community transmission and deaths [8,9]. The Indonesian government-imposed lockdowns known as Large-Scale Social Restrictions (Pembatasan Sosial Bersakal Besar/PSBB) in 2020 followed by the Community Activities Restrictions Enforcement (Pemberlakuan Pembatasan Kegiatan Masyarakat/PPKM) from early 2021 until May 2022 [10,11]. The lockdowns included closing public places, workplaces and schools, and restricting socio-cultural, religious, and public transport and community activities in public places or facilities [12,13].

The COVID-19 pandemic continues to cause a range of detrimental mental health impacts on different population groups including older people [14,15,16,17]. Mental health among older population refers to their state of well-being, which may include both their emotional or psychological well-being such as their feelings (happiness, feeling good with daily life activities and responsibilities, and feeling satisfied with life), and social well-being such as having good relationships or connections with friends or other community members, feeling part of their group or community, and having something to contribute to their society [18,19]. Older adults have an increased likelihood of acquiring SARS-CoV-2—the virus that causes COVID-19 infection [20]. Furthermore, older adults with COVID-19 infection have a disproportionally high risk of mortality, due to both age-related changes to the immune system and the presence of chronic diseases such as diabetes and cardiovascular disease [20,21]. Similarly, older adults with obesity are reported to experience severe impact of COVID-19 and increased risk of mortality due to impaired metabolic health [22,23]. For example, in the USA, older adults (>65 years) comprise 80% of all COVID-19-related deaths and 31% of COVID-19 infections, despite comprising 17% of the total population [20,24]. Moreover, the COVID-19 pandemic is uncovering the vulnerability of older adults, especially during national mandatory lockdown periods [25,26]. In high-income countries, recent research has revealed that older adults showed poorer mental health (anxiety and depression symptoms), reduced social functioning (increased isolation) and lower physical functioning compared with pre-pandemic levels [25]. Other studies have also reported that COVID-19 has caused them poor psychological health and well-being [27,28], fear of death, worry [27], stress [29], feeling of loneliness [30,31,32] and sleep disturbances which also led to increased depression [27,30].

Despite a range of COVID-19-related mental health challenges faced by different population groups, very little is known about the effects the COVID-19 pandemic’s mandatory lockdown protocols have on the daily life and mental health of older adults outside of high-income countries. Furthermore, in the context of Indonesia, there is a lack of study and evidence on the impacts of COVID-19 on older people. Several studies in the country which explored COVID-19-related mental health issues, such as depression, anxiety, stress, fear, and psychological distress, have mainly focused on healthcare professionals [33,34], university students [35,36], and the general population [37,38]. This qualitative study aims to fill in this gap by exploring in-depth the influence of COVID-19 outbreaks and the prevention protocols on individual and social activities of older people in Jakarta, Indonesia, and how such an influence impacts their mental health condition. Social activities refer to anything that brings older people together to interact with each other, as well as with other people, such as community meetings, group prayers, picnics, voluntary services at churches or Mosques, etc. [39]. Understanding these challenges facing older populations can be used to inform the development of community-based intervention programs that address unmet needs and help older populations in Indonesia and beyond cope with the challenges they face.

## 2. Methods

### 2.1. Study Design

A qualitative design using an in-depth interview method was employed [40]. A qualitative design was considered helpful in exploring the participants’ views, meanings, values and interpretations about their experience of personal and social activities and mental health challenges within the settings and situations where they lived and interacted [41]. This qualitative approach was effective in assisting the researcher to have a deeper understanding of the participants’ knowledge and experiences from their perspectives and contexts. It provided an insight into how older people made sense of their experiences and situations in relation to COVID-19-related mental health challenges [42].

### 2.2. Recruitment of Participants and Data Collection

This qualitative inquiry involved older people in Jakarta, Indonesia, to explore the impacts of the COVID-19 pandemic’s mandatory lockdown guidelines on their activities and mental health. Participants were recruited using the snowball sampling technique. Initially, the field researcher (E) purposively approached the leaders of churches and Mosques in the study setting and explained to them the purpose of the study and the possibility of recruiting older people within their communities. As the current data suggest that most Indonesian populations are affiliated with one of the religions in Indonesia, such as Islam, Protestant, Catholic, Hindu, or Buddhist, and only 0.04% do not have a religion, it is expected that older populations in the study setting have a religious affiliation [43]. Upon soliciting permission from them, the field researcher was allowed to post the study information sheets on their information boards. Potential participants who texted and confirmed their participation were recruited and asked to suggest a preferred time for an interview. At the end of the interviews, participants were asked to disseminate the information about the study to their colleagues or other older adults within their communities, who might be willing to participate. The use of the snowball sampling technique provided equal opportunity for eligible older adults in the study setting to participate in this study voluntarily. Older adults are basically defined as people who are over 60 years of age, but it is acknowledged that families and communities in many settings often define age category for older adults based on other sociocultural referents, such as family status [44]. In the context of Indonesia, older adults are defined as people aged 60 years or above [45]. Older adults who participated in this study were from different educational, work and religious backgrounds. They were recruited based on several inclusion criteria, including one having to be (i) an older person aged 60 or above, (ii) able to communicate well in Bahasa (Indonesian), and (iii) willing to participate voluntarily. Finally, a total of 24 older adults participated in this study (See Table 1). The age range of ≥60 was also chosen as one of the inclusion criteria based on the consideration of the average retirement age in Indonesia which starts at 58 years as stipulated in the government regulation [46]. The recruited participants were 60–82 years old, with the majority (*n* = 16) retired or not working, while the rest still engaged in some work. None of the participants were diagnosed with COVID-19 prior to the study and most had been vaccinated either once or twice or thrice (see Table 1).

Data collection was carried out using in-depth interviews during the pandemic from March to April 2022 when the Community Activities Restrictions Enforcement was still in place [47,48] and the COVID-19 cases were still rapidly increasing, from approximately 5.5 million cases in February 2022 to 6.04 million cases in April 2022 [49,50]. Due to the COVID-19 outbreak situation, interviews were conducted using online platforms: WhatsApp video calls or Zoom, which was mutually agreed upon by both participants and the interviewers (NKF, E). Before the interviews, participants were again advised that their participation was voluntary, and they had the right to withdraw at any time without any consequences. They also were assured information provided during the interviews will be used for publication purposes only and will be made anonymous and confidential. The duration of the interviews ranged from 30 to 55 min. Interviews were guided by several main research questions that focused on exploring the participants’ individual and social activities they regularly performed before the COVID-19 outbreak; their perceptions and experiences about the influence of the COVID-19 outbreak and the mandatory lockdown protocols on their daily individual activities; the influence of COVID-19 mandatory lockdown protocols on their social activities and social life, such as social connection with families, friends and other older adults; how COVID-19 pandemic’s lockdown protocols had influenced their mental health condition. All the interviews were recorded on WhatsApp or Zoom. The interviewer also took some notes during each interview once she felt necessary. None of the potential participants who had stated their intention to take part in this study withdrew their participation and no repeat interviews were carried out with any of the participants. Recruitment of participants and interviews ceased when the researchers felt that information/data obtained from the participants had been rich enough to address the research questions and objective and an indication of data saturation had been reached. Data saturation was indicated through the similarity of responses provided by the last few participants to those of previous participants. An informed consent which was sent to the participants via e-mail or WhatsApp a few days before the interviews was signed and returned to the researcher. All the interviews were carried out in Bahasa Indonesia (Indonesian national language), the primary language of the interviewers and the participants.

### 2.3. Data Analysis

Prior to the comprehensive data analysis, the recordings were transcribed verbatim and manually into coding sheets by two authors (NKF, GAA). Regular meeting and discussion of the codes and initial themes were undertaken between them throughout the analysis process. To retain the socio-cultural and religious meanings of the collected information, coding and analysis of the data were carried out in Indonesian and then selected quotes were translated into English (by NKF who is fluent in both Indonesian and English) for publication purposes [51]. To maintain the accuracy and clarity of the translation, we checked and rechecked the quotes against the translated interpretation to examine the meaning in both languages [52]. Team-based analysis and discussion about the themes and interpretations were conducted throughout the analysis and writing process through which comments and revisions were performed, and finally, all authors agreed on the final themes and interpretations presented in this manuscript. The analysis of the transcripts was carried out manually and guided by the five steps of qualitative data analysis introduced by Ritchie and Spencer in their framework analysis [53,54]. These steps include (i) familiarisation with the transcripts or data where each transcript was read thoroughly and repeatedly to be familiar with the content of the information provided by each participant. During the reading process, the transcript was broken down into chunks of information which enabled us to give comments and labels; (ii) identification of a thematic framework which was carried out by listing all the key concepts, recurrent issues, and ideas from the transcripts, which were then used to form the thematic framework; (iii) indexing the whole data. This was carried out by coding chunks of information in the transcripts, leading to a collection of open codes. These codes were then assessed to identify codes that were similar or redundant to reduce the number of codes (close coding). Codes that formed a logical pattern were then grouped into the same theme or sub-theme; (iv) charting the entire data. This was performed by reorganising and summarising all the themes and their codes created in the previous steps in a chart for comparison of the data within and across interviews; and (v) mapping and interpreting the data as a whole [54,55].

This study obtained ethics approval from the Medical and Health Research Ethics Committee, Krida Wacana Christian University, Jakarta, Indonesia (No. SLKE: 1242/SLKE-IM/UKKW/FKIK/KE/III/2022).

## 3. Results

### 3.1. Individual and Social Activities of Older Adults Prior to the COVID-19 Outbreak

Both male and female participants reported committing some common and regular individual activities before the COVID-19 outbreak. These included grocery shopping, doing regular outdoor sports in parks, visiting neighbours, visiting children and grandchildren, and going out with them to recreation places. Participants who were still working also reported actively engaging in their work-related activities such as catering, driving passengers, doing construction work, etc. For the majority of the older adults, engagement in such activities was seen as strategies to keep them physically active and strong, as a 71-year-old female participant said: “*I did my routine individual activities because I tried to be active and they kept my body strong*”. Similarly, another participant described ‘*Once you are getting old like me, the only way to keep your body blood circulation smooth in your body is physical activity* (Male, 69 years old). The stories of the participants also showed their awareness and understanding that being physically active is a healthy lifestyle that could make them not easily get sick and keep them away from diseases:

“*I always try to do as active as I can. Activities that I did before the COVID-19 outbreak were catering, visiting neighbours, and visiting children and grandchildren. Sometimes my children and grandchildren and I go to recreational places. I did all these activities to keep myself active and strong physically. I think that if I am physically active then the blood circulation in my body would be smooth, my immune system would be good and I might not get sick easily*”.(Female, 73 years old)

Participants also reported engaging in various social activities within their groups, communities, churches, and mosques prior to the COVID-19 outbreak. For example, they engaged in community meetings, neighbourhood association programs and activities, and regular social gatherings for older adults. One participant stated ‘*Before the COVID-19 outbreak there were regular meetings and activities within our community and I actively participated’* (Male, 69 years old). They also engaged in religious-related activities such as community prayer, Friday prayer, and Sunday mass, volunteered at the church, and visited places of the spiritual pilgrimage in groups. The stories of some of them illustrated that social and religious activities they engaged in had positive meanings and values for them, as reflected in the following quotes:

“*I was actively involved in social activities in our group and community before the COVID-19 outbreak. Doing small things together and for others made me feel useful. Also, serving and talking to people at the church are good things to do, I was at the church every week before COVID came* …”.(Female, 60 years old)

*I try to fill my life with good things, that is why I was involved in community prayer, visiting pilgrimaging spiritual places, and helping others. These are things that show how valuable you are to other people* …”.(Male, 75 years old)

Some participants also described that engaging in social activities with other people within groups or communities and religious groups helped them stay busy and connected to other people and avoid getting stuck at home which could lead to negative thoughts or stress and anxiety as illustrated in the following narrative:

“*I think older people like me should keep themselves busy with various activities and connect with other people within their communities or groups. Before the spread of COVID-19, I was involved in various social activities at the group, neighbourhood, and community levels. I was also very active in the church and helped minister the Sunday services. So, every week I always had something to do. These were not meaningless activities. These activities made me feel happy and I was not stuck at home. Doing these activities regularly helped me avoid negative thoughts, stress, or anxiety. Older people like me sometimes have the thoughts that they would die soon or are not physically strong anymore and are vulnerable to many health problems. I don’t want such thoughts to go through my head and scare me, that’s why I was involved in various activities*”.(Female, 60 years old)

### 3.2. The Impacts of the COVID-19 Pandemic’s Mandatory Lockdown Protocols on the Activities of Older Adults

COVID-19’s mandatory lockdown protocols, such as social/physical distancing and avoiding crowded places, were reported as having a significant influence on both individual and social activities of male and female older adults in this study. Some described how these prevented them from doing their activities and kept them staying at home for months. For example, a 64-year-old female participant commented: “*I could not go anywhere and just stayed at home due to this COVID-19 outbreak*”. Similarly, another participant stated, “*I could not even go to the market because of the mandatory lockdown protocols which suggest us to avoid crowded place and also the fear of COVID-19 transmission”* (Female, 60 years old). The fear of contracting COVID-19 and breaching the prevention protocols was described by the participants as the main underlying reasons for their adherence to the lockdown protocols:

“*I am personally scared of contracting corona, that is why I just stayed at home for more than a year. I am still scared but now I go out to supermarkets sometimes. The government put the COVID-19′s prevention protocols into practice when the number of cases was rising, and as an older person, I must obey these protocols because I am much more vulnerable to the infection and the impacts could be severe if I am infected*”.(Male, 62 years old)

Participants who were still actively working raised the same comments suggesting that COVID-19’s mandatory lockdown protocols significantly influenced their work-related activities. For example, those who worked as motor taxi drivers, Bajaj drivers, construction workers, and catering service providers described that they could not work due to a lack of demand from clients or passengers during the COVID-19 outbreak and when the mandatory lockdown protocols were put in practice. The following story from a motor taxi driver reflects such an influence:

“*I did not work at all for one year and a few months because it seemed that people were scared of travelling using a motorbike taxi. People were scared of contracting COVID-19… Even now people are still scared of going with a motor taxi, scared of sitting close to the driver on a motorbike*”.(Male, 61 years old)

Similarly, the participants also described that COVID-19’s mandatory lockdown protocols had a significant influence on their social activities. For example, some described that the protocols had led to the cancelation of their social and religious activities such as social gatherings and picnics with other older adults, community prayers, and meetings. Such an influence is reflected in the following narratives of the participants:

“*When COVID-19 pandemic’s mandatory lockdown protocols were put in place, activities such as group prayers and community meetings were cancelled. The pilgrimage activities that we did regularly have also been cancelled until now due to the COVID-19 protocols*”.(Female, 71 years old)

“*In our community, there were social gatherings for parents and older people and we sometimes went to picnics together. Since the COVID-19 outbreak and due to the enforcement of lockdown protocols we have stopped social gatherings and picnics until now. Moreover, the information I heard was that older people like me are more vulnerable to COVID-19 and if they contract it then the consequences can be fatal, so these social activities have been cancelled until now*”.(Male, 69 years old)

Some participants also emphasised how COVID-19’s mandatory lockdown protocols had led to their social disconnection with other people such as friends and people whom they cared about and who cared about them. Such feeling of social disconnection seemed to be supported by their perception and experience that they needed people their age to share experiences and engage in conversation as reflected in the following story of a 75-year-old man:

“*The enforcement of COVID-19’s mandatory lockdown protocols disconnects me from my friends. As an old person, I think it is bad to get disconnected from people my age. I need people my age, my friends to talk to because we understand each other* …”.

### 3.3. The Impacts of the COVID-19 Pandemic’s Mandatory Lockdown Protocols on the Mental Health of Older Adults

#### 3.3.1. Feelings of Loneliness and Social Disconnection

The COVID-19 outbreak and the enforcement of lockdown protocols by the government also negatively impacted the mental health of several female and male older adults who participated in this study. The feeling of loneliness was one of the mental health challenges they faced during the outbreak. They described how the COVID-19 outbreak and the mandatory lockdown protocols had led to them spending most of their time alone and doing nothing at home. A 70-year-old female participant described:

“*I have been spending most of the time alone at home during this COVID-19 outbreak. I hardly go out of my house; the mandatory protocols are clear and I also am scared of this COVID. One of my children lives with me but he goes to work every day*”.

The experience of the feeling of loneliness seemed to be even worse among older adults who lived alone, as is illustrated in the following quote:

“*This COVID-19 situation has been the worst experience in my life. I live alone, I am on my own, you can imagine. There are times I felt so lonely. Before the COVID-19 outbreak started and the enforcement of mandatory lockdown protocols, I could go anywhere at any time I wanted because I am retired. …. I don’t like this situation at all*”.(Male, 69 years old)

Some participants also described that spending most of their time at home for a long period due to the COVID-19 pandemic’ mandatory lockdown protocols caused them strange feelings or feeling imprisoned. One of the participants explained: “*It feels so strange during the COVID-19 pandemic, I feel like I am imprisoned at home. Even now the number of COVID cases keep increasing and we cannot go anywhere because the protocols are still in place*” (Male, 68 years old). Engaging in the same activities or daily routine in their homes and having limited space for their movement were clearly expressed in their stories as the supporting factors for their experience of such feelings as reflected in the following narrative:

“*One of the things that makes me feel very bored is doing the same things every day in my house. …. My movements are mainly inside the house and this has been going on for quite a while. It feels like I am being imprisoned in my own house*”.(Female, 74 years old)

Social disconnection was another strong supporting factor for these negative feelings experienced by older adults in this study. It was apparent from their stories that the COVID-19 pandemic’s mandatory protocols disrupted their social lives and connections. This was reflected in their narratives showing that they were unable to meet and visit other people including their older adult friends with their groups or communities due to the outbreak and the related restrictions:

“*Before the COVID-19 outbreak and the enforcement lockdown protocols, I was always with my friends, Bajaj drivers. We always wait for passengers together at the same spot but since the beginning of the COVID-19 outbreak and then the government put the lockdown protocols in place, we didn’t meet each other. …. I feel that there is a big and sudden change, we are disconnected with each other up to now*”.(Male, 69 years old)

“*I have some close friends, we often meet either during community activities or at church. We also often go together. But due to COVID-19 lockdown protocols, we have not met each other in person..... There is a sense of loneliness and boredom because this situation has been going on for quite a long time, more than two years*”.(Female, 76 years old)

#### 3.3.2. Feelings of Loss

Feelings of loss were another mental health consequence experienced by participants during the COVID-19 outbreak. Some participants shared their feeling of loss related to social events or activities they engaged in regularly, which they stopped doing due to the mandatory protocols for COVID-19 prevention. For example, a 73-year-old woman who was actively involved in Sunday mass services in her church described:

“*I feel like something has been missing in my life. It is probably because I do not do any more things (catering and Sunday service at the church) that I regularly did before the mandatory lockdown protocols take effect. I stopped since the early stage of the COVID-19 outbreak*”.

A few participants also raised their comments about the deaths of their friends (older adults) and their inability to visit due to COVID-19’s mandatory lockdown protocols as the causes of the feeling of loss they experienced:

“*I have experienced the loss of people who I knew and were close to me during the COVID-19 pandemic. There have been some of my friends and relatives who died but I couldn’t attend their funerals because of the COVID-19 protocols. …. Before the COVID-19 outbreak, we met every week, shared stories, prayed together, and so on but we lost contact with each other during the pandemic. I feel lost*”.(Female, 71 years old)

Similar stories were also echoed by three participants who experienced the loss of their relatives and extended family members due to COVID-19 infection and other health issues during the pandemic. The deaths of members of families were described as causing them a great experience of loss. Such an experience seemed to be hurtful for them as they were not allowed by COVID-19 protocols to attend the funerals. The following narrative of a male participant who lost a relative and nephew during the pandemic reflects the feeling of loss the participants experienced:

“*I lost two family members due to COVID-19 infection; one was my relative who was the same age as me and another one was my nephew who was still young. The sad thing was that I couldn’t go to see them or attend their funerals. Their bodies and funerals were taken care of or handled by health workers or people who have been trained*”.(69 years old)

#### 3.3.3. Feeling Stressed, Worried, Sad, and Broken

Feelings of stress and worry were other common mental health issues the participants faced during the pandemic. Some participants described that the whole COVID-19-related situation, including the enforcement of the mandatory lockdown protocols, sometimes made them feel stressed and worried. Besides, some described those feelings of stress and worry sometimes made it difficult for them to sleep:

*“…. New COVID cases are always reported on TV or radio and this makes me feel worried. I feel like the situation compels me to do things not in the usual ways; wearing a facemask, social distancing, avoiding crowded places, and so on. These stress me out. If I went out and met people, I became worried about contracting COVID and then I couldn’t sleep during the night*”.(Male, 67 years old)

Feeling sad and broken were also mental health impacts experienced by some participants. The reason was that the enforcement of COVID-19’smandatory lockdown protocols had a negative influence on their economic condition, which caused mental health challenges for them. A participant stated, “*I was made redundant since the first COVID-19 outbreak and the lockdowns. This makes me feel confused and sad because I have no work to do”* (Male, 65 years old). Other participants who engaged in several jobs such as catering, motor taxi drivers, Bajaj drivers, and construction workers described how the COVID-19’s lockdown protocols had forced them to quit their jobs or income-generating activities due to a lack of demand or order. However, quitting these jobs was acknowledged to make them feel sad and broken as it led to their inability to provide for their families. The following narratives of a 73-year-old widow who used to do catering and a 65-year-old man who worked as a Bajaj driver clearly show the mechanisms of how the COVID-19 situation influenced their financial and mental health conditions:

“*I used to do small catering for a few years before the COVID-19 outbreak. My regular customer was XX church. I did the catering for most of the events at the church but since the COVID-19 outbreak and the enforcement of the mandatory lockdown protocols, there were no Sunday mass and meetings. All activities at the church stopped to comply with COVID-19 protocols. It influences me financially up to now and I feel sad and a bit broke because it is very difficult to provide for myself and the needs of my family*”.

“*Since COVID-19’s mandatory lockdown protocols were put in place I stopped driving Bajaj because there were no passengers. Sadly, I had to quit the job, and even more sad is the fact that sometimes I could not have enough money to feel secure enough and fulfill the needs of my family*”.

## 4. Discussion

This study provides novel evidence on the influence of COVID-19 lockdowns on individual and social activities in older adults, and how these negatively impacted the mental health of community-dwelling older people in Indonesia. Older adults represent a population group at very high risk of adverse mental health outcomes due to social isolation, particularly during the lockdown periods during the COVID-19 pandemic [25,56]. However, very little research has focused on the effect of COVID-19 lockdowns on older adults. The preponderance of previous studies that explored its impacts has mainly focused on the mental health consequences it had on healthcare providers, students or young people, adult women and men in general, and infected people [14,16,57]. Similarly, some COVID-19 studies involving older people have mainly explored mental health challenges facing them [27,28,29,30,58]. To our knowledge, our study is one of the first to focus on the daily functioning of older adults in low-middle income countries. The findings from our study both agree and disagree with that of recent literature from high-income countries. For instance, research from the Netherlands reported that the mental health of community-dwelling older adults was scarcely affected by mandatory lockdown periods during the first COVID-19 waves [59], with the authors of this study concluding that older adults either had better coping strategies than younger adults, or that daily lifestyle changes were minimal. Similar findings were also reported in studies in other European countries [60], the USA and Canada [61,62] suggesting that older people had less negative mental health effects or a lower rate of poor mental health conditions (anxiety, depression, stress) and more often reported positive daily events compared to younger adults.

In our study, it is believed that mental health was detrimentally affected in older adults due to the fear of widespread COVID-19 infection and awareness of their vulnerability to the infection and its severe negative impacts on their health. However, such fear and awareness seemed to support the participants’ compliance with COVID-19’s lockdown protocols. Our findings reflect the concepts of perceived susceptibility and severity which suggest that people with high perceived susceptibility to contracting an infection, and who believe in the seriousness of an infection and its sequelae, are highly likely to perform recommended behaviours (e.g., in this case maintaining social/physical distancing and avoiding crowded places) [63,64,65]. It is also apparent in our data that there was a disruption of the daily functioning and social life of these older adults caused by the enforcement of the COVID-19 pandemic’s mandatory protocols.

Previous studies have found that the COVID-19 outbreak and its prevention protocols contributed to mental health challenges, such as the feeling of loneliness, in older people in different settings [31,32]. Such experience was also identified in the older adults who participated in our study. More than that, our study offers an in-depth understanding of the mechanism through which COVID-19 and its prevention protocols contributed to such mental health issues among older populations. Our findings also note that the social activities of the older adults have social meanings and values (e.g., being useful to other people), and facilitate their social relationships and networks. For some, such meanings, values, relationships, and networks were the sources of positive and happy feelings. Thus, the disruption of their social activities also contributed to social disconnection and the feeling of loneliness among them. These are consistent with the findings of some previous non-COVID-related studies which have reported that social disconnection puts older people at greater risk for morbidity and mortality, and has significant effects on their mental health conditions [66,67].

Our study also reported the feelings of loss experienced by participants due to the deaths of families and friends during the COVID-19 pandemic, which have not been reported in previous COVID-related studies with older populations in other settings [27,28,29,58]. Such experience seemed to be exacerbated by COVID-19’s mandatory lockdown protocols in the context of Indonesia which do not allow people, other than COVID-trained staff, to visit or take part in funerals of people who died from COVID-19 [68].

Other studies have previously reported the significant negative influence of the COVID-19 pandemic’s mandatory prevention protocols on psychological health and well-being [27,28], stress and anxiety [29] in older people. Some of these mental health issues, including stress, worry, sadness, and feeling broken were also apparent in our data. What our study adds to the existing knowledge on this notion is the influence of COVID-19’s mandatory lockdown protocols on older people’s economic or financial condition. For some participants who engaged in food catering, Bajaj or motor taxi driving, and construction work, the mandatory lockdown protocols had led to the cancelation of social events (from where they received food catering orders) and a lack of passengers and customers. These not only led to the participants experiencing loss of income but also quitting their job, and being unable to provide for their families, which were acknowledged as contributing to the feeling of sadness, broken, and stress.

The current study involved a small number of participants; hence the findings mainly reflect the perspectives and experiences of older people we interviewed or do not represent the general views and experiences of older adults in other parts of Indonesia. Thus, similar to most qualitative studies, the findings of this study are not meant to be generalised to the older populations in Indonesia and globally. Similarly, it involved both older adult who were retired and actively engaging some work which seemed to lead to different views and experiences about the influence of lockdowns on their work or financial conditions. Data collection method through online platforms was also another possible limitation of the study as it may have led us to under-sampling older people who did not possess a smartphone or computer, who may have different stories and experiences of the impact of COVID-19 on their activities, economic and mental health conditions. However, our findings have significant implications for the government of Indonesia at the national, provincial, and district levels to address the impacts of COVID-19 on the mental health and economic conditions of older people at policy and practical levels. The large number of older adults negatively affected by social isolation and inactivity during COVID-19 lockdown periods should make this population group a priority. Older adults at risk of social isolation should be provided with social connection material (for instance, telephone, email or WhatsApp) delivered online or over the phone by trained, volunteer workers. Potentially, such input could be provided as part of a national social prescribing program. Similarly, older adults experienced economic difficulties should be continually supported financially. Although, the government of Indonesia provided monthly cash social assistance of IDR 300,000 (±UDS 20) to 10 million Indonesian families, this program did not target older adults and ceased in the beginning of 2021 [69,70]. Our findings provide useful information to inform the development of intervention programs that address the need of older adults and support them in coping with those impacts. Future studies involving large-scale older populations to comprehensively understand COVID-19 impacts on them are recommended. Future studies involving older adults with disability and comorbidities to understand different views and experiences of COVID-19 related impact on older adults are also recommended.

## 5. Conclusions

This study presents the influence of the COVID-19 pandemic’s lockdown protocols on the individual and social activities of older adults in Jakarta, Indonesia, which further affected their mental health. Daily functioning through individual and social activities was acknowledged as keeping them healthy and strong physically and mentally. Similarly, their social activities facilitated social relationships and networks among them and between them and other people and were considered as having social meaning and values that made them happy and felt useful to others. Thus, disruption of their daily functioning due to the COVID-19 pandemic’s mandatory lockdown protocols negatively affected their mental health or contributed to the feelings of loneliness, loss, stress, worry, sadness, and social disconnection among them. As the older adults’ awareness of their vulnerability to COVID-19 infection and its related risk that could contribute to poor their mental health and well-being, the findings suggest that there is a need for tailored intervention programs that address the socio-economic and mental health impacts of the COVID-19 pandemic on this population to help them cope with these challenges.

## Figures and Tables

**Table 1 ijerph-19-13115-t001:** Characteristics of the participants.

Characteristics	No = 24
Age	
60–65	9
66–70	4
71–75	6
≥76	5
**Sex**	
Female	13
Male	11
**Education**	
Junior high school	6
Senior high school	10
Diploma/Bachelor	8
**Employment**	
Retired/Not working	16
Drivers (motorbike taxi and bajaj)	2
Self-employed (catering and culinary business)	2
Private Employees	3
Construction workers	1
**Living Situation**	
Living with spouse, children and grandchildren	16
Living alone with a spouse	3
Living alone with a brother or sister	2
Living alone	3
**Vaccine Status**	
Not vaccinated	4
Vaccinated once	9
Vaccinated twice	6
Vaccinated thrice	5

## Data Availability

The data collected in this study are available in the manuscript and on request from the corresponding author.

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
