# Peer review of "Impact of COVID-19 Lockdowns on the Activity and Mental Health of Older People in Indonesia: A Qualitative Study"

_ijerph, 2022, doi:10.3390/ijerph192013115_

Round 1
Reviewer 1 Report
Dear authors,
thank you, I've read your manuscript with expectations. Association between COVID-19, lockdowns and mental health status is an interesting topic, worth tackling. Although your paper has potential it needs many improvements.
I have some questions and comments. Your paper is very subjective and „one side views only“. You can not make generalisations out of it (line 462).
The methodology is quite weak. Please add some more information, if it is possible:
Characteristic of the population, clearly describe whom you asked- why did you choose people 60 years old and over, not 65 and up? The definition of the older adult is missing. When do people retire in your country and receive their pension? All older people from your sample have financial difficulties during lockdowns? (line 20-21) Add inclusion and exclusion criteria.
Better characteristics of individuals -e.g. their overall objective health status before/after the pandemic (your study), did they suffer from COVID-19 - the severity of disease, hospitalization, their vaccination status, as well as their family and friends (21/24 people were living with their close family, so they were not totally separated, is not it?), what about other stressors in their overall life?...and think about a lot of other modifiers.
Characteristic of Lockdowns-what was forbidden, what was allowed to do?
Characteristic COVID-19 situation during your study (when it was conducted)? Because it could influence your participants a lot, if they already felt „safe“ (which wave was running, increasing-decreasing direction, how many people have been affected during that time? Among their friends and family? Was it forbidden to go to nature, outside?...)
Line 62- obesity as the main risk factor is missing
Line 467-How do you distinguish what is a consequence of lockdown, or due to fear or stress of getting COVID or long Covid, on mental health changes?
Line 177-218 - all activities are listed, that only lucky-healthy elderly people could do, what about disabled, polymorbid -the majority of the elderly population?
There are only 3 systematic reviews used, it would be helpful to do a deeper search.
Author Response
Reviewer 1
Comment: I have some questions and comments. Your paper is very subjective and “one side views only“. You cannot make generalisations out of it (line 462).
Response: The current study involved a small number of participants; hence the findings mainly reflect the perspectives and experiences of older people we interviewed or do not represent the general views and experiences of older adults in other parts of Indonesia. Thus, similar to most qualitative studies, the findings of this study are not meant to be generalised to the older population in Indonesia and globally.
Comments: The methodology is quite weak. Please add some more information, if it is possible. Characteristic of the population, clearly describe whom you asked- why did you choose people 60 years old and over, not 65 and up? The definition of the older adult is missing. When do people retire in your country and receive their pension? All older people from your sample have financial difficulties during lockdowns? (line 20-21) Add inclusion and exclusion criteria. Better characteristics of individuals -e.g. their overall objective health status before/after the pandemic (your study), did they suffer from COVID-19 - the severity of disease, hospitalization, their vaccination status, as well as their family and friends (21/24 people were living with their close family, so they were not totally separated, is not it?), what about other stressors in their overall life?...and think about a lot of other modifiers.
Responses: In terms of more details on methods, we have the following in the manuscript –
“The use of the snowball sampling technique provided equal opportunity for eligible older adults in the study setting to participate in this study voluntarily. Older adults are basically defined as people who are over 60 years of age, but it is acknowledged that families and communities in many settings often define age category for older adults based on other sociocultural referents, such as family status [42]. In the context of Indonesia, older adults are defined as people aged 60 years or above [43]. Older adults who participated in this study were from different educational, work and religious backgrounds. They were recruited based on several inclusion criteria, including one having to be (i) an older person aged 60 or above, (ii) able to communicate well in Bahasa (Indonesian), and (iii) willing to participate voluntarily. Finally, a total of 24 older adults participated in this study (See Table 1). The age range of ≥60 was also chosen as one of the inclusion criteria based on the consideration of the average retirement age in Indonesia which starts at 58 years as stipulated in the government regulation [44]. The recruited participants were 60-82 years old, with the majority (n=16) retired or not working, while the rest still engaged in some work. None of the participants were diagnosed with COVID-19 prior to the study and most had been vaccinated either once or twice or thrice (see Table 1).”
In terms of the financial condition of participants, only a few shared the stories related to financial difficulties which also contributed to mental health challenges facing them as reflected in the result section:
“Feeling sad and broken were also mental health impacts experienced by some participants. The reason was that the enforcement of COVID-19’smandatory lockdown protocols had a negative influence on their economic condition, which caused mental health challenges for them. A participant stated, “I was made redundant since the first COVID-19 outbreak and the lockdowns. This makes me feel confused and sad because I have no work to do” (Male, 65 years old). Other participants who engaged in several jobs such as catering, motor taxi drivers, Bajaj drivers, and construction workers described how the COVID-19’s lockdown protocols had forced them to quit their jobs or income-generating activities due to a lack of demand or order. However, quitting these jobs was acknowledged to make them feel sad and broken as it led to their inability to provide for their families. The following narratives of a 73-year-old widow who used to do catering and a 65-year-old man who worked as a Bajaj driver clearly show the mechanisms of how the COVID-19 situation influenced their financial and mental health conditions.”
In terms of the overall objective health status before/after the pandemic: As this was qualitative study carried out at one period of time, we did not record the participants health status before and after the pandemic.
In terms of other stressors or modifiers: We would like to emphasise that our study focused on exploring participants views and experiences of the impact of COVID-19 lockdowns on their activities and mental health. We acknowledge that there may be other stressors or modifiers which are out of our focus in this study.
Comment: Characteristic of Lockdowns-what was forbidden, what was allowed to do?
Response: Thank you for the comment, these are the ones stipulated in the Government Regulation No 21, 2020 (in the manuscript)
The Indonesian government-imposed lockdowns known as Large-Scale Social Restrictions (Pembatasan Sosial Bersakal Besar/ PSBB) in 2020 followed by the Community Activities Restrictions Enforcement (Pemberlakuan Pembatasan Kegiatan Masyarakat/PPKM) from early 2021 until May 2022 [10, 11]. The lockdowns included closing public places, workplaces and schools, and restricting socio-cultural, religious, and public transport and community activities in public places or facilities [12, 13].
Comment: Characteristic COVID-19 situation during your study (when it was conducted)? Because it could influence your participants a lot, if they already felt “safe“ (which wave was running, increasing-decreasing direction, how many people have been affected during that time? Among their friends and family? Was it forbidden to go to nature, outside?...)
Response: The following is in the manuscript:
Data collection was carried out using in-depth interviews during the pandemic from March to April 2022 when the Community Activities Restrictions Enforcement was still in place [10, 11] and the COVID-19 cases were still rapidly increasing, from approximately 5.5 million cases in February 2022 to 6.04 million cases in April 2022 [44, 45].
Comment: Line 62- obesity as the main risk factor is missing
Response: The following is in the manuscript:
Furthermore, older adults with COVID-19 infection have a disproportionally high risk of mortality, due to both age-related changes to the immune system and the presence of chronic diseases such as diabetes and cardiovascular disease [20, 21]. Similarly, older adults with obesity are reported to experience severe impact of COVID-19 and increased risk of mortality due to impaired metabolic health [22, 23].
Comment: Line 467-How do you distinguish what is a consequence of lockdown, or due to fear or stress of getting COVID or long Covid, on mental health changes?
Response: Thank you for the comment. We would like to emphasise that lockdowns were implemented or socialised together with the message about the possibility of contracting COVID-19 and its negative on health and life, which often caused fear, stress, worry and mental health changes on older adults in this study. These aspects were interrelated in the current participants’ experiences.
Comment: Line 177-218 - all activities are listed, that only lucky-healthy elderly people could do, what about disabled, polymorbid - the majority of the elderly population?
Response: Thank you for raising this comment. Unfortunately, none of participants had these health conditions, thus the narratives about the activities do not represent the stories and experiences of these groups. However, this is an important point to be explored further in future studies.
Comment: There are only 3 systematic reviews used, it would be helpful to do a deeper search.
Response: Additional resources have been added throughout the manuscript.
Reviewer 2 Report
First of all, I want to congratulate the authors for the study. In fact, the article reports an important study for the academic and scientific community. The study presents scientific and methodological criteria. It follows the ethical and deontological principles of scientific research. After some changes, which are described later, it must be accepted for publication.Suggested changes in the article:
Line 48-50 – Are there no other references that can be included, namely from scientific publications?
Line 383 – reference (s) used?
Author Response
Reviewer 2
Comment: Line 48-50 – Are there no other references that can be included, namely from scientific publications?
Response: We have cited relevant journal articles and government regulations.
Comment: Line 383 – reference (s) used?
Response: References have been added.
Reviewer 3 Report
This is a nice contribution to the qualitative literature on the impact of COVID-19 lockdowns on mental well-being among older adults. It provides an important perspective from an LMIC which is a valuable contribution to the literature. I have some fairly minor suggestions to help further improve the paper:
Editorial suggestions:
- On Line 32: Chinese Centres for Disease Control should be capitalized
- One line 36: "leads" should be "led"
- On line 53: The definition of mental health is confusing as it sounds like all mental health related to older adults. I suggest revising it to: "Mental health among older populations refers to..."
- On line 86 and 105: There is inconsistency in the capitalization of "Mosques"
- On line 108: I believe "Buddha" should be "Buddhist"
Other Comments:
- In the Abstract and Methods section, please clarify that you used a Framework Analysis approach to qualitative analysis. This is mentioned and the process is well-described in the methods but it is not introduced clearly and is not mentioned in the abstract.
- It would be helpful to provide more detail in the Methods section, including:
1) who conducted the analysis- what is one or more people? If and how were consensus reached on the themes that were identified?
2) please specify whether the interviews were structured, semi-structured, etc.
3) Who translated the transcripts to English and at what stage did this occur?
- In the results section, please separate the quotations from the main text using paragraph indent and be consistent in the formatting. For example: "Quote" (Sex, age)
- In section 3.3 the authors use sub-headings to distinguish the themes. It would be helpful to use sub-headings throughout the Results section to improve clarity
- In the Discussion section, the authors frequently refer to the findings providing insight into cause of poor mental health. Since this is a qualitative study, it does indeed provide in-depth understanding for factors contributing to poor mental among older adults. I would caution against the use of "cause" however, as this isn't an epidemiological study. Instead I suggest using words like "contributes to".
- In the Discussion on Line 424 I suggest adding a paragraph break for clarity as a new topic is introduced.
- Throughout the Discussion, if the word limit allows, it would be helpful to include more detail about how this study relates to findings from other regions or countries and some further interpretation of the implications of these findings. While references are provided for other studies, it would strengthen this section if the authors engaged more with the broader literature, including factors that might help to explain differences or similarities in findings across different studies.
- The authors discuss several implications of the results for consideration by government to better support the mental well-being of older adults during the pandemic or future emergencies. I would encourage them to add considerations for financial support given the impact of the pandemic on their income and its relationship to poor mental well-being. Did older adults in Indonesia receive any social assistance during the pandemic? Was this available to people who lost their income during the pandemic? What would the authors recommend?
- Finally, in the Conclusion the authors state: "As the older populations appear to be vulnerable to COVID-19 infection that could adversely impact mental health and well-being..." This sentence is confusing and appears to suggest that COVID-19 infection itself could adversely impact mental health and well-being. While this might be the case, I believe that the authors actually mean that older adults' awareness of their vulnerability to COVID-19 infection and it's related risks contributes to poor mental well-being.
Author Response
Reviewer 3
Comment: On Line 32: Chinese Centres for Disease Control should be capitalized
Response: Done
Comment: One line 36: "leads" should be "led"
Response: Done
Comment: On line 53: The definition of mental health is confusing as it sounds like all mental health related to older adults. I suggest revising it to: "Mental health among older populations refers to..."
Response: Done
Comment: On line 86 and 105: There is inconsistency in the capitalization of “Mosques”
Response: Revised: ‘M’ is used.
Comment: On line 108: I believe “Buddha” should be “Buddhist”
Response: Buddhist is used.
Comment: In the Abstract and Methods section, please clarify that you used a Framework Analysis approach to qualitative analysis. This is mentioned and the process is well-described in the methods but it is not introduced clearly and is not mentioned in the abstract.
Response: Data analysis was guided by the five steps proposed in a qualitative data analysis framework, including familiarisation with the data, identification of a thematic framework, indexing the data, charting the data and mapping and interpreting the data.
Comment: It would be helpful to provide more detail in the Methods section, including: 1) who conducted the analysis- what is one or more people? If and how were consensus reached on the themes that were identified? Who translated the transcripts to English and at what stage did this occur?
Response: This is in the manuscript:
Prior to the comprehensive data analysis, the recordings were transcribed verbatim and manually into coding sheets by two authors (NKF, GAA). Regular meeting and discussion of the codes and initial themes were undertaken between them throughout the analysis process. To retain the socio-cultural and religious meanings of the collected information, coding and analysis of the data were carried out in Indonesian and then selected quotes were translated into English for publication purposes [51]]. To maintain the accuracy and clarity of the translation, we checked and rechecked the quotes against the translated interpretation to examine the meaning in both languages [52]]. Team-based analysis and discussion about the themes and interpretations were conducted throughout the analysis and writing process through which comments and revisions were performed, and finally, all authors agreed on the final themes and interpretations presented in this manuscript.
Comment: please specify whether the interviews were structured, semi-structured, etc.
Response: This is in the manuscript:
Interviews were guided by several main research questions that focused on exploring the participants’ individual and social activities they regularly performed before the COVID-19 outbreak; their perceptions and experiences about the influence of the COVID-19 outbreak and the mandatory lockdown protocols on their daily individual activities; the influence of COVID-19 mandatory lockdown protocols on their social activities and social life, such as social connection with families, friends and other older adults; how COVID-19 pandemic’s lockdown protocols had influenced their mental health condition.
Comment: In the results section, please separate the quotations from the main text using paragraph indent and be consistent in the formatting. For example: "Quote" (Sex, age)
Response: We have done so. As you can now see, only some very short quotes are place within main text to support our synthesis as it is common in qualitative papers. The long ones are separated from them main text.
Comment: In section 3.3 the authors use sub-headings to distinguish the themes. It would be helpful to use sub-headings throughout the Results section to improve clarity
Response: Thank you for the comment. We do not use sub-heading in sections 3.1 and 3.2 as these are very short themes and straightforward compared to section 3.3 which is a big theme that needs to be elaborated and separated in sub-heading for clarity and ease to follow.
Comment: In the Discussion section, the authors frequently refer to the findings providing insight into cause of poor mental health. Since this is a qualitative study, it does indeed provide in-depth understanding for factors contributing to poor mental among older adults. I would caution against the use of "cause" however, as this isn't an epidemiological study. Instead I suggest using words like "contributes to".
Response: The revision has been made throughout the discussion section.
Comment: In the Discussion on Line 424 I suggest adding a paragraph break for clarity as a new topic is introduced.
Response: Done
Comment: Throughout the Discussion, if the word limit allows, it would be helpful to include more detail about how this study relates to findings from other regions or countries and some further interpretation of the implications of these findings. While references are provided for other studies, it would strengthen this section if the authors engaged more with the broader literature, including factors that might help to explain differences or similarities in findings across different studies.
Response: Thank you for the comment, we have provided references for studies in other settings globally and for concepts related to the topic, linked the current findings to the previous findings and discussed what our study adds to the previous findings or existing knowledge on the topic.
Comment: The authors discuss several implications of the results for consideration by government to better support the mental well-being of older adults during the pandemic or future emergencies. I would encourage them to add considerations for financial support given the impact of the pandemic on their income and its relationship to poor mental well-being. Did older adults in Indonesia receive any social assistance during the pandemic? Was this available to people who lost their income during the pandemic? What would the authors recommend?
Response: The following is in the manuscript:
Similarly, older adults experienced economic difficulties should be continually supported financially. Although, the government of Indonesia provided monthly cash social assistance of IDR 300,000 (± UDS 20) to 10 million Indonesian families, this program did not target older adults and ceased in the beginning of 2021 [69, 70].
Comment: Finally, in the Conclusion the authors state: "As the older populations appear to be vulnerable to COVID-19 infection that could adversely impact mental health and well-being..." This sentence is confusing and appears to suggest that COVID-19 infection itself could adversely impact mental health and well-being. While this might be the case, I believe that the authors actually mean that older adults' awareness of their vulnerability to COVID-19 infection and it's related risks contributes to poor mental well-being.
Response: Thanks for the detailed comment, this has been revised as follows:
As the older adults’ awareness of their vulnerability to COVID-19 infection and its related risk that could contribute to poor their mental health and well-being, the findings suggest that there is a need for tailored intervention programs that address the socio-economic and mental health impacts of the COVID-19 pandemic on this population to help them cope with these challenges.
Reviewer 4 Report
Thank you for the opportunity to review your paper. I agree it is an important paper to share perspectives of the people of Indonesia and the comparison to higher-income countries. I have some minor suggestions for clarity of the paper, otherwise, I felt it was well written with clear and sound methodology.
1. Line 14, consider adding the word "social" before activities.
2. Line 121 & 134, consider capitalizing Zoom
3. Consider sharing direct information about Indonesia's lockdown protocol at the time if that is available. The information shared is more general to other countries/world.
4. Line 277 (3.3.1), consider stating how many participants expressed this feeling. There were quantifying words used within 3.3.2 ("some") and also in 3.3.3
Author Response
Reviewer 4
Comment: Line 14, consider adding the word "social" before activities.
Response: Done
Comment: Line 121 & 134, consider capitalizing Zoom
Response: Done
Comment: Consider sharing direct information about Indonesia's lockdown protocol at the time if that is available. The information shared is more general to other countries/world.
Response: Thank you for the comment, these are the ones stipulated in the Government Regulation No 21, 2020:
The Indonesian government-imposed lockdowns known as Large-Scale Social Restrictions (Pembatasan Sosial Bersakal Besar/ PSBB) in 2020 followed by the Community Activities Restrictions Enforcement (Pemberlakuan Pembatasan Kegiatan Masyarakat/PPKM) from early 2021 until May 2022 [10, 11]. The lockdowns included closing public places, workplaces and schools, and restricting socio-cultural, religious, and public transport and community activities in public places or facilities [12, 13].
Comment: Line 277 (3.3.1), consider stating how many participants expressed this feeling. There were quantifying words used within 3.3.2 ("some") and also in 3.3.3
Response: Done
Reviewer 5 Report
I would like to thank the authors for their work.
This is an interesting paper, which aims to to explore the impacts of the COVID-19 pandemic’s man-13 datory lockdown protocols on activities and mental health conditions of community-dwelling older 14 people in Jakarta, Indonesia.
I would suggest some minor revisions:
1) the choice of this age cut-of may result uncommon in some countries; please motivate this choice
2) the sample includes some workers, leading to some potential differences compared to retirees; I would highlight this aspect between the limits of the study
Author Response
Reviewer 5
Comment: the choice of this age cut-of may result uncommon in some countries; please motivate this choice
Response: The following is in the manuscript:
Older adults are basically defined as people who are over 60 years of age, but it is acknowledged that families and communities in many settings often define age category for older adults based on other sociocultural referents, such as family status [44]. In the context of Indonesia, older adults are defined as people aged 60 years or above [45]. ………. The age range of ≥60 was also chosen as one of the inclusion criteria based on the consideration of the average retirement age in Indonesia which starts at 58 years as stipulated in the government regulation [46].
Comment: the sample includes some workers, leading to some potential differences compared to retirees; I would highlight this aspect between the limits of the study
Response: The current study involved a small number of participants; hence the findings mainly reflect the perspectives and experiences of older people we interviewed or do not represent the general views and experiences of older adults in other parts of Indonesia. Thus, similar to most qualitative studies, the findings of this study are not meant to be generalised to the older populations in Indonesia and globally. Similarly, it involved both older adult who were retired and actively engaging some work which seemed to lead to different views and experiences about the influence of lockdowns on their work or financial conditions.
Round 2
Reviewer 1 Report
Thank you for accepting comments.